# Tension Force Stress Downregulates the Expression of Osteogenic Markers and Mineralization in Embryonic Stem-Cell-Derived Embryoid Bodies

**DOI:** 10.3390/cells14130991

**Published:** 2025-06-28

**Authors:** Ju-Hyeon An, Chun-Choo Kim, Junil Lee, Junhyeok Kim, Jeong-Chae Lee, Sung-Ho Kook

**Affiliations:** 1Cluster for Craniofacial Development and Regeneration Research, Institute of Oral Biosciences, School of Dentistry, Jeonbuk National University, Jeonju 54896, Republic of Korea; lovedent28@gmail.com (J.-H.A.); chunsams@hanmail.net (C.-C.K.); uhaha147@gmail.com (J.L.); km5442e@naver.com (J.K.); 2Department of Bioactive Material Sciences, Research Center of Bioactive Materials, Jeonbuk National University, Jeonju 54896, Republic of Korea

**Keywords:** tensile stress, embryoid body, extracellular signal-regulated kinase, osteogenic marker, mineralization

## Abstract

Mechanical stresses affect a variety of cellular events in relation to the frequency, magnitude, and duration of the stimuli applied. Embryonic stem cell (ESC)-derived embryoid bodies (EBs) are pluripotent stem cell aggregates and comprise all somatic cells. Numerous studies have highlighted the effects of mechanosignals on stem cells, whereas their impact on EBs has been barely investigated. Here, we examined how cyclic tensile stress affects the behavior of EBs to differentiate into mineralized osteocytes by applying 2% elongation at 0.5 Hz frequency for 1 h once or 1 h every other day for 5 or 14 days in osteogenic medium. EBs that expressed undifferentiated markers, Oct4 and Sox2, were differentiated into mineralized cells, along with the accumulation of runt-related transcription factor 2 (RUNX2) and β-catenin in osteogenic medium. The application of tensile force inhibited EB’ mineralization via the downregulation of bone sialoprotein, osteocalcin, osterix, and RUNX2. While the transfection with si-β-catenin did not affect the osteogenic potency of EBs at a significant level, treatment with 10 μM of PD98059, but not of SP600125 or SB203580, diminished the mineralization of EBs and the expression of RUNX2 and RUNX2-regulated osteoblastic genes. The level of phosphorylated extracellular signal-regulated kinase-1 (p-ERK1) rather than p-ERK2 was more apparently diminished in tension-applied EBs. The transfection with si-ERK1, but not with si-ERK2, suppressed the mineralization of osteogenic medium-supplied EBs and the expression of osteoblast-specific genes. Collectively, this study demonstrates that tensile stress inhibits osteogenic potency of EBs by downregulating ERK1-mediated signaling and osteogenic gene expression.

## 1. Introduction

Embryoid bodies (EBs) are the 3D aggregates of embryonic stem cells (ESCs) and enable differentiation into various lineage cells [1]. The culture of EBs allows the production of pluripotent stem cells on a large scale and extends their clinical usefulness in regenerative medicines [1,2]. Considerable findings also indicate that the application of EBs in combination with the extracellular matrix (ECM) provides great advantages in guided tissue regeneration (GTR) [3,4,5]. ECM components are pivotal physiological regulators for cellular differentiation through integrin-derived signaling activation and can provide cues to modulate the expansion and differentiation of ESCs and/or EBs into a specific lineage. Regarding this, the scaffolds with native bone ECM components can directly enhance osteogenic differentiation of ESCs and bone formation.

While EBs are an important source of uncommitted progenitor cells and exert beneficial effects in GTR, it is difficult to control the behaviors of EBs during the cultures within a guided 3D scaffold [6]. Among various factors that modulate cell behaviors, mechanical stimuli are the key mediators that affect the proliferation, differentiation, and migration of cells [7,8,9]. Scaffolds themselves may activate various mechanosignals and affect the behaviors of cells in relation to their fiber topographies [10,11]. A strain stress is the main mechanical stimulus that is exposed to cells grown on or within scaffolds and modulates the fate of cells to proliferate and/or differentiate into functionalized cells [12]. However, the effects of strain stress on the osteogenic differentiation and mineralization of EBs and the associated mechanisms are barely understood. Therefore, the understanding of the mechanosignal mechanisms by which strain stress modulates the osteogenic behavior of ESC-derived EBs is necessary to improve their clinical approach for GTR.

Of the various mechanosensing molecules, mitogen-activated protein kinases (MAPKs) such as extracellular signal-regulated kinase (ERK), c-Jun N-terminal kinase (JNK), and p38 MAPK play crucial roles in the mechanosignal transduction [13,14]. The activated MAPKs transfer exogenous mechanosignals into the nucleus and mediate transcriptional activation depending on the conditions of mechanical stimuli. The Wnt/β-catenin signaling is also activated by mechanosignals and regulates the differentiation of cells into various lineage cells [15]. Accordingly, our aims of this study are focused on exploring how strain stress affects the osteogenic potential of EBs and investigating which of β-catenin or MAPK acts as the main mechanosignal mediator in the stress-applied EBs. Here, we produced homogeneous EBs from ESCs using a hanging drop method and applied them to 2% cyclic tensile stress using a computer-controlled tension equipment. This study demonstrates that tensile loading stress inhibits the osteogenic differentiation and mineralization of EBs via the downregulation of ERK1/RUNX2-regulated signaling.

## 2. Materials and Methods

### 2.1. Cells, Chemicals, and Laboratory Wares

The mouse ES cell line D3 was purchased from the American Type Culture Collection (Rockville, MD, USA). Fetal bovine serum (FBS) was purchased from HyClone Laboratories, Inc. (Logan, UT, USA), while Dulbecco’s modified Eagle’s medium (DMEM) was from Thermo Fisher Scientific (Waltham, MA, USA). Pharmacological MAPK inhibitors specific to JNK (SP600125), ERK (PD98059), and p38 kinase (SB203580) were obtained from Tocris Bioscience (Bristol, UK). Monoclonal or polyclonal antibodies specific to β-actin (sc-7199), β-catenin (sc-515105), ERK (sc-94), p-ERK (sc-7383), JNK (sc-6245), p38 kinase (sc-535), runt-related transcription factor 2 (RUNX2, sc-10758), and α-tubulin (sc-8035) were obtained from Santa Cruz Biotechnology, Inc. (Dallas, TX, USA). Alizarin Red S (ARS; TMS-008-C) was purchased from Sigma-Aldrich Co. LLC (St. Louis, MO, USA). BioFlex^®^ plates that are type I collagen (COLI)-coated and flexible-bottomed 6-well plates were purchased from FlexCell International Corporation (Burlington, NC, USA). Unless specified otherwise, other antibodies, chemicals, and laboratory consumables were obtained from Sigma-Aldrich Co. LLC or SPL Life Sciences (Pochun, Republic of Korea).

### 2.2. Cell Culture and EB Production

ESCs were grown in DMEM including 15% FBS, 1.7 mM l-glutamine, 0.1 mM β-mercaptoethanol, 5 ng/mL mouse leukemia inhibitory factor (LIF), 3.7 g/L sodium bicarbonate, and antibiotics (1% penicillin and streptomycin) without a feeder layer under the conditions of 37 °C and 5% CO_2_. EBs were produced from ESCs using a handing drop method. Briefly, when the number of ESCs reached 90% confluence in culture plates, cells were disaggregated in a trypsin-EDTA buffer and resuspended in LIF-free growth medium. The medium drops (25 μL) containing approximately 3000 ESCs/drop were plated on the lid of 100 mm culture plates in regular arrays, and the lids were inverted and placed over the bottom of the plates. The culture plates containing hanging drops were incubated for 2 days before the addition of LIF-free growth medium (10 mL/plate). To induce the formation of EBs, the cultures were incubated for 3 additional days in a body floating condition. EBs more than 300 μm in diameter were harvested and randomly divided onto the COLI-coated flexible-bottomed or commonly sterilized 6-well plates. After 24 h of the seeding, the plates filled with the same batch of growth medium or osteogenic medium supplemented with 100 nM dexamethasone, 40 μM ascorbic acid, and 5 mM β-glycerophosphate (DAG). After an additional 24 h of incubation, EBs located on the flexible plates were applied to tensile stress, whereas the EBs in the sterilized plates received siRNA transfection or MAPK inhibitor treatment followed by further assays at various post-incubation times.

### 2.3. Immunofluorescence Assay

The nature of EBs to express undifferentiated and osteogenic-specific markers was determined in the cultures supplemented with and without DAG by immunofluorescence staining. Briefly, EBs were fixed with 2% paraformaldehyde solution and washed three times with phosphate-buffered saline (PBS). EBs were incubated with 0.25% Triton X-100 and washed with PBS containing 0.05% Tween^®^ detergent prior to the treatment with bovine serum albumin. After washing and blocking processes, EBs were exposed to primary Oct4, Sox2, RUNX2, or osterix antibody. Appropriately coupled secondary antibodies such as Alexa Fluor 488- and 594-conjugated anti-IgG (Abcam, Cambridge, UK) were used for single and double labeling. After the counterstaining with DAPI, the expression patterns of Oct4, Sox2, RUNX2, and osterix were evaluated under a confocal laser scanning microscope (CLSM; Carl Zeiss, Oberkochen, Germany).

### 2.4. Application of Tensile Stress

The bottom of the flexible culture plates was subjected to static waves using a computer-controlled vacuum stretch equipment (FX-4000^TM^ Tension Plus^TM^, FlexCell International Cooperation, Burlington, NC, USA), through which EBs were applied to cyclic tensile stress (2% elongation of 0.5 Hz frequency) for 1 h once or 1 h every other day for 5 or 14 days.

### 2.5. siRNA Transfection and Pharmacological Inhibition of MAPKs

EBs were transfected with si-β-catenin, si-ERK1, si-ERK2, or both si-ERK1 and 2 with 25-bp Stealth^TM^ Select RNAi (HSS102353, Invitrogen, Carlsbad, CA, USA). The coding strand for β-catenin or ERK(s) siRNA was a duplex of the sequences as follows: uac aau ggc aga cac cau cug agg g-antisense, ccc uca gau ggu guc ugc cau ugu a-sense for β-catenin; aaa ggu uaa cau ccg guc cag cag c-antisense, ccu gcu gga ccg gau guu aac cuu u-sense for ERK1; and ugu cga acu uga aug gug cuu cgg c-antisense, gcc gaa gca cca uuc aag uuc gac a-sense for ERK2. The transfection of EBs with siRNAs was performed using Lipofectamine™ 2000 (Invitrogen) according to the manufacturer’s instructions. The EBs transfected with an unrelated siRNA were used as the negative control. Portions of EBs were also treated with 10 μM of SP600125, PD98059, or SB203580 and incubated for 14 days followed by the analyses of mineralization and expression of osteogenic regulatory molecules. Cultures were replaced with the same batch of medium every 3 days during the incubation.

### 2.6. Mineralization Assay

ARS staining was performed to evaluate the osteogenic potential of EBs. Briefly, at the end of incubation, EBs were fixed with 70% ethanol on ice for 1 h and treated with 0.2% ARS dissolved in distilled water at 25 °C for 30 min. After washing twice, ARS-stained EBs were observed under a camera-installed light microscope (Nikon TS100, Tokyo, Japan). The absorbance of ARS-specific dye was also determined by extracting the dye from EBs using 10% acetylpyridinum chloride followed by the measurement of optical density at 405 nm. In addition, the number of EBs exhibiting more than 300 μm in diameter and their average sizes (μm in diameter) were measured after ARS staining.

### 2.7. Polymerase Chain Reaction (PCR) Assay

The mRNA expression of alkaline phosphatase (ALP), bone sialoprotein (BSP), osteocalcin (OCN), osteopontin (OPN), osterix, and RUNX2 in EBs was determined by real-time PCR. To this end, total RNA was extracted from EBs using TRIzol reagent according to the manufacturer’s instructions (Invitrogen). The cDNAs were synthesized from RNA extracts (1 μg/sample) using an AmpiGene cDNA kit (Enzo Life Sciences, Inc., Farmingdale, NY, USA) following the manufacturer’s instructions. The PCR was performed using an ABI 7500 sequence detection system (Applied Biosystems, Foster City, CA, USA), and the accumulation of PCR products was monitored using Power SYBR Green PCR Master Mix (Applied Biosystems). In this assay, PCR products were amplified for 40 cycles under the conditions of denaturation at 95 °C for 15 s, annealing at 60 °C for 1 s, and extension at 72 °C for 30 s after predenaturation at 95 °C for 10 min. The mRNA expression of RUNX2 in siRNA-transfected EBs was also evaluated using an Access PCR System (Promega, Madison, WI, USA). The reverse transcription and amplification of RNA samples were performed using a DNA thermal cycler (PTC-100, MJ Research Inc., Waltham, MA, USA) as described previously [13]. PCR products for *RUNX2* and glyceraldehyde 3-phosphate dehydrogenase (*GAPDH*) were analyzed by agarose gel electrophoresis, after which a gel imaging system (BIO-RAD, Segrate, Italy) was used to determine band intensities. Oligonucleotide primers of osteogenic molecules were designed as follows: forward (F) 5′-ggg ccc tgc tgc ttc cac tg-3′, reverse (R) 5′-ggc ttg tgg gac cct gca ccc-3′ for *ALP*; F 5′-gag gga cta tgg cgt caa aca-3′, R 5′-gga tcc caa aag aag ctt tgc-3′ for *RUNX2*; F 5′-tca gcc gcc ccg atc ttc ca-3′, R 5′-aat ggg tcc acc gcg cca ag-3′ for *osterix*; F 5′-tgg tgg tga tct agt ggt gcc aa-3′, R 5′-cac cgg gag gga gga ggc aa-3′ for *OPN*; F 5′-act ccg gcg cta cct tgg gt-3′, R 5′-cct gca gtc tag ccc tct gc-3′ for *OCN*; F 5′-aga cca gga ggc gga ggc ag-3′, R 5′-ttg ggc agt tgg agt gcc gc-3′ for *BSP*; and F 5′-gac ggc cgc atc ttc ttg t-3′, R 5′-cac acc gac ctt cac cat ttt-3′ for *GAPDH*. In this study, *GAPDH* was used as the endogenous internal marker.

### 2.8. Preparation of Protein Lysates

Whole, cytosolic, and nuclear protein lysates were extracted and quantified as described previously [16]. Briefly, EBs were harvested at various times (0–48 h) after the first application of tensile stress, and whole protein extracts were prepared using an NP-40 lysis buffer (pH 7.5) supplemented with 1 mM EDTA, 150 mM NaCl, 1% NP-40, and 30 mM Tris-Cl. For cytosolic fractionation, EBs were incubated in 150 μL buffer A (pH 7.5) containing 1 mM dithiothreitol, 1 mM EDTA, 1 mM EGTA, 20 mM Hepes, 10 mM KCl, 1.5 mM MgCl_2_, 1 mM phenylmethylsulfonyl fluoride, 250 mM sucrose, and 10 μg/mL each of aprotinin, leupeptin, and pepstatin A on ice for 30 min. Cells were resuspended 20 times using a 26 G needle before centrifugation at 750× *g* for 10 min. Supernatants were collected and centrifuged at 10,000× *g* for 25 min. Thereafter, the aqueous layers were harvested as cytosolic protein fraction, whereas the pellets were used as nuclear fractions after several processes for additional resuspension 20 times in buffer A (100 μL) and centrifugation at 10,000× *g* for 25 min.

### 2.9. Western Blotting

Sodium dodecyl sulfate-polyacrylamide gel electrophoresis (10–12%) was performed to separate whole or fractioned protein extracts (30 μg/extract), followed by blotting onto polyvinyl difluoride membranes. Blots were washed with 10 mM Tris-HCl buffer (pH 7.6) supplemented with 150 mM NaCl and 0.05% Tween-20 before treatment with 5% skim milk for 1 h. The blots were probed with each of the primary antibodies (1:200 dilution) at 4 °C for 12 h and exposed to secondary antibodies. Blots were treated with enhanced chemiluminescence (ELPIS-Biotech, Daejeon, Republic of Korea) in a dark room and then exposed to X-ray film (Eastman Kodak, Rochester, NY, USA). The intensities of immunoreactive bands were determined using the ImageJ densitometry program (National Institutes of Health, Bethesda, MD, USA) after normalizing them to that of control markers such as β-actin, ERK, and α-tubulin.

### 2.10. Statistical Analysis

All data are expressed as the mean ± standard deviations. Significant differences between the two groups were analyzed by unpaired Student’s *t*-test using GraphPad Prism software 9 (Boston, MA, USA). A value of *p* < 0.05 was considered statistically significant.

## 3. Results

### 3.1. Experimental Design and Characterization of Undifferentiated and Osteogenic Marker Expression in EBs in the Presence and Absence of DAG

Figure 1 illustrates the experimental designs for the production of EBs from ESCs, the seeding of EBs onto flexible-bottomed six-well plates, and the application of cyclic tensile stress using the FLEXERCELL tension equipment. We firstly determined the levels of pluripotency-related markers in EBs at several time points of incubation. Representative staining images of undifferentiated markers in EBs are shown in Figure 2A. Oct4 and Sox2 were strongly expressed in 1-day EBs, and that expression was found in 3-day EBs. The 7-day EBs also showed the expression level of Sox2 similar to that in 1-day EBs, but the level of Oct4 in 7-day EBs was significantly reduced (Figure 2B). We next examined the property of EBs to express osteogenic transcription factors, RUNX2 and osterix, in the cultures supplemented with or without DAG. The immunofluorescence staining revealed that EBs on day 3 expressed RUNX2 and osterix, and that expression was obviously increased by the treatment with DAG (Figure 2C,D).

### 3.2. Cyclic Tensile Stress Diminishes DAG-Induced Mineralization of EBs, and This Is Orchestrated by the Downregulation of Osteogenic Regulatory Molecules

We examined how the tensile stress affects DAG-induced mineralization of EBs by ARS staining. Compared with EB cultures in growth medium, DAG-supplemented EBs showed higher numbers of ARS-positive colonies, and this increase was diminished in the EBs applied to 2% tensile stress (Figure 3A). The measurement of ARS-specific optical density (405 nm) supported the DAG-induced mineralization of EBs and its significant decrease (*p* < 0.05) by tensile stress (Figure 3B). The tensile stress-applied cultures also revealed significantly lower numbers (*p* < 0.05) of EBs exhibiting more than 300 μm in diameter compared with those supplemented with DAG only (Figure 3C). In addition, DAG-supplemented EBs expressed significantly greater levels (*p* < 0.001) of *RUNX2*, *osterix*, *BSP*, and *OCN*, but not of *ALP* and *OPN*, compared with the growth-medium-supplied EBs (control group, Figure 3D). DAG-stimulated expression of these osteogenic genes in EBs was significantly (*p* < 0.05 or *p* < 0.01) attenuated by the combination with tensile stress. Similarly, immunoblot assay showed the DAG-stimulated induction of RUNX2 and its suppression by tensile force (Figure 3E,F).

### 3.3. DAG-Stimulated Mineralization of EBs and Its Inhibition by Tensile Stress Are Not Directly Affected by β-Catenin-Mediated Signaling

To understand why tensile stress suppresses DAG-stimulated mineralization of EBs, we initially checked the protein levels of integrins and β-catenin in EBs. Compared with growth-medium-supplied control EBs, DAG-treated EBs showed apparent increases in integrins α5 and β1 and β-catenin in whole protein lysates, and these increases were not changed by tensile stress at the indicated times (0–48 h) after the tensile loading (Figure 4A). The band intensities specific to integrins and β-catenin in DAG-supplied EBs were also significantly (*p* < 0.05 or *p* < 0.01) higher compared with those in the control group, and these increases were not diminished after the tensile stress (Figure 4B–D). Additionally, the EB groups supplemented with DAG only or in combination with tensile stress showed significantly greater levels of β-catenin both in cytosolic (*p* < 0.01) and nucleic factions (*p* < 0.001) than did the control group (Figure 4E–G).

To explore the role of β-catenin-mediated signaling on osteogenic differentiation of DAG-stimulated EBs, we transfected them with si-control or si-β-catenin. Immunoblot assay showed that the β-catenin level in whole protein lysates was significantly (*p* < 0.05) diminished by si-β-catenin transfection (Figure 5A,B). Compared with the EBs supplemented with DAG or in combination with si-control transfection at 14 days post-incubation, ARS intensity in the si-β-catenin-transfected EBs tended to be slightly diminished (Figure 5C). However, the mean absorbance of ARS dye (Figure 5D) and the number of ARS-positive colonies (Figure 5E) in the si-β-catenin-transfected EBs did not differ with them treated with DAG only or together with si-control transfection at a significant level.

### 3.4. Tensile Stress Diminishes Phosphorylation of ERK1 in DAG-Stimulated EBs

To evaluate the effect of ERK-mediated signaling on tension-applied EBs, the levels of p-ERK were compared among the control, DAG, and DAG/tension-applied EB groups by immunoblot assay (Figure 6A). An immediate and temporal increase of p-ERK was found in tension-applied EBs compared with the control or DAG group; however, the DAG-stimulated phosphorylation of ERK was extensively suppressed after 12 h of tensile stress. Densitometric analysis supported that the tension-mediated decrease of p-ERK in DAG-supplied EBs was further prominent in p-ERK1 rather than p-ERK2 (Figure 6B). These results indicate that the ERK1-mediated signaling is closely associated with the tension-mediated anti-osteogenesis in DAG-stimulated EBs.

### 3.5. Pharmacological ERK Inhibitor Attenuates DAG-Mediated Mineralization of EBs via the Downregulation of Osteogenic Marker Genes

We further investigated the effects of MAPKs on the DAG-stimulated mineralization of EBs by treating with each (10 μM) of MAPK inhibitors. The results from ARS staining indicated that DAG-induced mineralization of EBs was suppressed by the pharmacological inhibitor of p-ERK, but not of p-JNK or p-p38 kinase (Figure 7A). Among the MAPK-treated EB groups, a significant decrease (*p* < 0.05) in the ARS dye-specific absorbance was found only in the PD98059-exposed EBs compared with the DAG control EBs (Figure 7B). The pretreatment with PD98059 also significantly (*p* < 0.05) inhibited the DAG-stimulated expression of *RUNX2*, *osterix*, *BSP*, and *OCN* in EBs (Figure 7C). In addition, the DAG-mediated induction of RUNX2 and osterix proteins in 5-day EBs was significantly (*p* < 0.05) reduced in the combined treatment with the pharmacological ERK inhibitor (Figure 7D,E).

### 3.6. ERK1-Mediated Signaling Acts as an Upstream Effector of RUNX2 in DAG-Stimulated EBs

To clarify the roles of ERK-mediated signaling in the DAG-stimulated mineralization of EBs, we transfected the bodies with si-ERK1, si-ERK2, or both. The DAG-induced mineralization of EBs was attenuated by the transfection with si-ERK1 rather than with si-ERK2 (Figure 8A). Measurement of the ARS dye-specific optical density confirmed the association of ERK1-mediated signaling with the DAG-stimulated mineralization of EBs, such that si-ERK1-transfected EBs revealed significantly lower mineralization (*p* < 0.01) compared with DAG control group (Figure 8B). The tensile stress-mediated decrease in mineralization of DAG-stimulated EBs was closely associated with their circular sizes, in which the average colony diameter of si-ERK1- or si-ERK1/2-transfected EBs was significantly (*p* < 0.05) smaller than that of the DAG control group (Figure 8C). The transfection with si-ERK1 or si-ERK1/2, but not with si-control or si-ERK2, significantly (*p* < 0.05) suppressed the levels of *RUNX2* (Figure 8D) and *OCN* (Figure 8E) in DAG-supplied EBs. Furthermore, the result from agarose gel electrophoresis indicated that ERK1- rather than ERK2-mediated signaling mediates the mineralization of DAG-stimulated EBs by regulating the expression of RUNX2 (Figure 8F,G). Taken together, the current findings suggest that supplemental DAG mineralizes EBs via the upregulation of ERK1/RUNX2-associated signaling, whereas tensile stress inhibits DAG-induced calcification of EBs by downregulating ERK1-mediated signaling (Figure 8H).

## 4. Discussion

Mechanical signals not only modulate the behaviors of cells consisting of bone tissues [17] but also affect cellular properties to differentiate into various lineage cells [18,19]. When cells are exposed to a mechanical stimulus, they convert the mechanosignals to cellular signals, transfer the signals into the nucleus, and then activate the expression of mechanosignal-responsive genes [9]. While understanding the mechanisms by which stem cells receive and transfer mechanosignals and activate mechanosensing molecules is important in stem-cell-applied bone repair, the role of tensile-stress-derived mechanosignals on the osteogenic potential of EBs has barely been investigated. It is also important to consider that tensile stress stimulates mechanosignal-mediated cellular events differently in relation to monolayered cells or their aggregates, the supplementation of growth, or a differentiating medium, as well as the conditions of mechanical stimuli [20]. We previously found that tensile stress at 5% or 10% elongation for 1 h induces β-catenin-associated cardiomyogenesis in EBs via cellular redox-associated and integrin/Akt-stimulated pathways [21]. However, we also found that the application of cyclic tensile force for 5 days with more than 4% elongation destroys the circularity of EBs. Moreover, the application of tensile force inhibited the proliferation of periodontal fibroblasts by upregulating cell cycle inhibitory signaling depending on the magnitude of elongation [22]. Regarding this, this study selected 2% elongation at 0.5 Hz as the tensile stress.

There are numerous molecules that are reactive to exogenous mechanosignals. The β-catenin transduces Wnt-associated signaling, and the Wnt/β-catenin signaling pathway plays important roles in cell adhesion and gene transcription [9,23]. The β-catenin regulates a variety of mechanosignal-associated cellular responses [24,25]. Integrins play critical roles in mechanosignal transduction by directly interacting with the ECM and the actin cytoskeleton. Integrin-induced mechanosignal transduction is an important process for the development and homeostasis of tissues [25]. Integrin complexes also play pivotal roles in transferring mechanical cues into the cytoplasm in response to tensile stress [26]. This study explored whether integrins and β-catenin play a role in the tensile-stress-mediated suppression of DAG-induced EB’ mineralization. Our results indicated that in addition to integrins α5 and β1, β-catenin signaling does not directly affect DAG-stimulated mineralization of EBs and its inhibition by tensile stress, despite its cytosolic and nuclear accumulations in DAG-supplied EBs. Our results are controversial with respect to the findings showing that strain stress promotes osteogenic differentiation of stem cells [27]. Together, we assume that Wnt/β-catenin signaling is not directly associated with tensile-stress-mediated anti-osteogenesis in DAG-stimulated EBs. Furthermore, we consider that integrin-associated mechanotransduction does not influence the DAG-stimulated mineralization of EBs, although additional experiments are necessary.

Otherwise, MAPKs are known to regulate mechanosignal-mediated cellular behaviors in various types of cells [28]. MAPK-mediated signaling is involved in the tension-mediated regulation of osteogenic and osteoclastic responses in cells. Dissimilar to the previous studies [13,14,28], our current findings show that tension stress causes a temporal activation of ERK, whereas the p-ERK level is extensively diminished in an incubation-time-dependent manner. As proven by the results showing that the p-ERK, but not p-JNK and p-p38 kinase, inhibitor diminishes the mineralization of DAG-stimulated EBs, our findings indicate the regulatory role of ERK-stimulated signaling in DAG-exposed EBs. Furthermore, the results from si-ERK transfection support the critical role of p-ERK1 rather than p-ERK2 in the DAG-induced mineralization of EBs. Taken as a whole, our results suggest that tension-derived mechanical signals inhibit the activation of ERKs in DAG-supplied EBs, in which ERK1-mediated signaling tightly affects the osteogenic potential of the bodies.

The differentiation of cells into mineralized osteocytes is orchestrated by the induction of various osteogenic molecules in a differentiation-stage-dependent manner. Among the molecules, RUNX2 is a master regulator of the processes required for osteoblast differentiation, bone matrix gene expression, and mineralization [29,30]. RUNX2-regulated osteogenesis is accompanied by the expression of its downstream molecules, including osterix, ALP, BSP, COLI, OPN, and OCN. The induction of these molecules is tightly dependent on the stages of osteogenic differentiation, in which OCN is expressed at a relatively late stage [31]. Our results suggest that ERK1-mediated signaling acts as the upstream effector of RUNX2 and RUNX2-regulated osteogenic molecules in DAG-stimulated EBs. Our findings also indicate that unlike osterix, BSP, and OCN, the expression of ALP and OPN is not altered by supplemental DAG and/or tensile stress. While the basal activity of ALP varies in relation to the origins of cells, its activity in ESCs is highly maintained regardless of the differentiated stages [32]. In addition to the role of OPN as a bone component, it also acts as a crucial regulatory component in the niches of fetal bone marrow stem cells [33]. In addition, OPN has important roles in modulating the behaviors of stem cells through the CXCR4 signaling axis [34]. These properties of ALP and OPN may disturb their application as the usual markers of osteogenic differentiation in DAG-stimulated and/or tension-applied EBs, although their roles in DAG-supplied EBs remain questionable. Collectively, our results postulate that tensile force inhibits the osteogenic potency of EBs via the ERK1-mediated downregulation of RUNX2 and RUNX2-associated downstream molecules.

Investigating the mechanical signaling networks that occur between cells and scaffolds is important in understanding cell behaviors in relation to the fiber topographies of scaffolds. Here, we have tried to understand the effect of tensile stress on the mineralization of DAG-exposed EBs and the associated mechanisms therein. Our current findings provide scientific information on the roles of tensile-stress-derived mechanosignals on the osteogenic potential of EBs. Overall, this study may suggest a meaningful consideration that the loading of EBs into scaffolds that express relatively high tensile strength disturbs EB-derived synergistic enhancement in the scaffold-mediated healing of defective bones. However, it is important to note that the elongation-induced tensile stress might affect cell behaviors differently from that derived from the scaffold structure-derived tensile strength. Additional experiments using EBs loaded onto 3D-structured fibrous scaffolds will be needed to verify their actual properties in response to tensile force, as well as to improve their clinical usefulness for GTR.

## 5. Conclusions

This study examined the effect of cyclic tensile stress on the osteogenic potency of DAG-stimulated EBs and the associated mechanisms. This study demonstrates that tensile stress suppresses DAG-stimulated osteogenic differentiation and mineralization of EBs. Our results also suggest that ERK1-mediated signaling, but not Wnt/β-catenin signaling, is closely associated with the tension-mediated anti-osteogenesis in DAG-stimulated EBs. Overall, our findings indicate the role of p-ERK1 as the upstream effector of RUNX2 in DAG-stimulated EBs.

## Figures and Tables

**Figure 1 cells-14-00991-f001:**
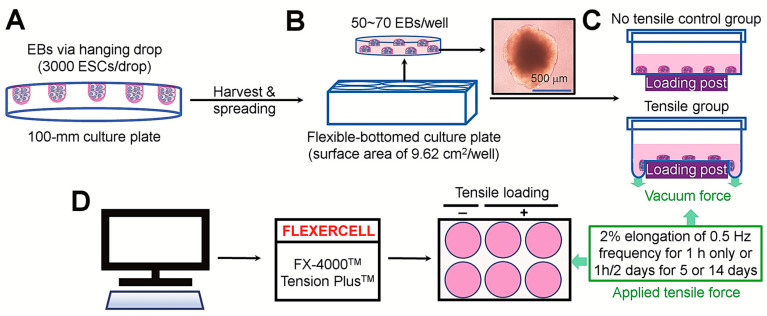
Experimental designs for tensile stress application to ESC-derived EBs. (**A**) EBs were produced from ESCs, (**B**) seeded on flexible-bottomed plates, and (**C**) applied to 2% elongation of 0.5 Hz frequency for 1 h once or 1 h every other day for 5 or 14 days in the presence and absence of DAG using (**D**) the FEXERCELL equipment.

**Figure 2 cells-14-00991-f002:**
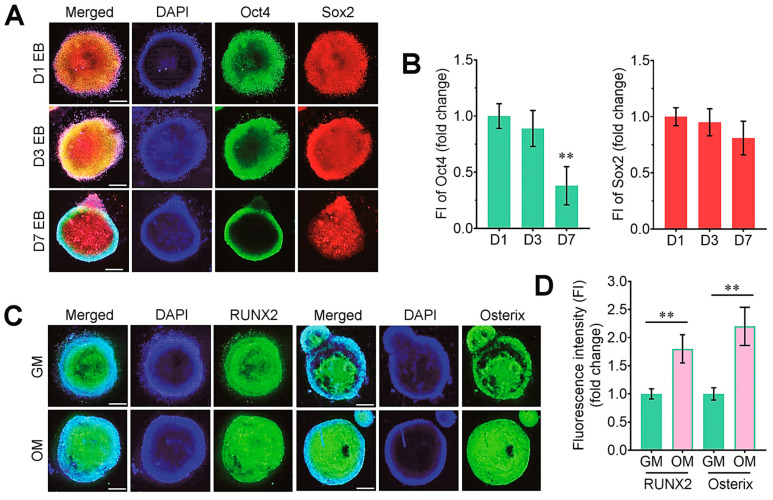
Immunofluorescence staining of undifferentiated and osteogenic markers in EBs. (**A**) The levels of Oct4 (green) and Sox2 (red) were evaluated in 1-, 3-, and 7-day EBs. (**B**) Fluorescence intensity (FI) of these markers was compared in relation to the incubation days. (**C**) The levels of RUNX2 and osterix proteins were determined in 3-day EBs supplemented with DAG (OM) or not (GM) by immunostaining assay. (**D**) Levels of these osteogenic transcription factors in the EBs were compared in relation to those cultures. ** *p* < 0.01 vs. the D1 or between the two groups by unpaired Student’s *t*-test (*n* = 5). Bar = 100 μm. GM, growth medium; OM, osteogenic medium.

**Figure 3 cells-14-00991-f003:**
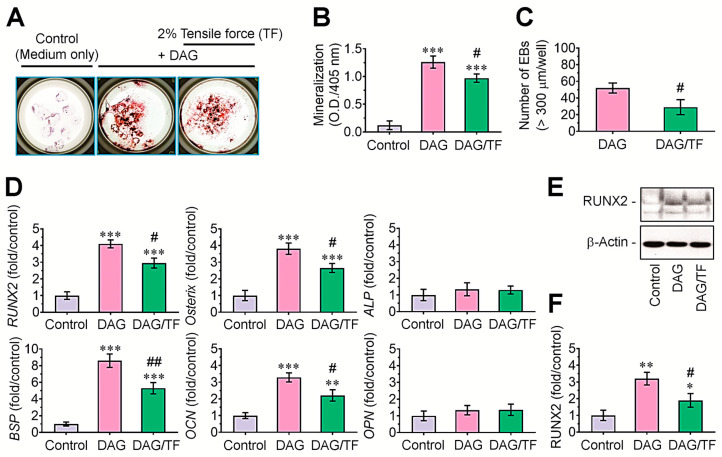
Cyclic tensile stress inhibited osteoblastic differentiation and osteogenic marker expression in DAG-stimulated EBs. (**A**) EBs were incubated in growth medium or DAG-supplied osteogenic medium with and without tensile stress, and at 14 days post-incubation, ARS-stained bodies were photographed under optic microscope. (**B**) The absorbance specific to the ARS dye and (**C**) the number of EBs more than 300 μm in diameter were determined. (**D**) The levels of *RUNX2*, *osterix*, *BSP*, *OCN*, *ALP*, and *OPN* were analyzed by real-time RT-PCR at 5 days post-incubation. (**E**) Protein level of RUNX2 and (**F**) its fold change were determined by Western blotting at the same post-incubation time after normalizing the band intensity to that of β-actin. A representative result from three different samples is shown. * *p* < 0.05, ** *p* < 0.01, and *** *p* < 0.001 vs. the control by unpaired Student’s *t*-test (*n* = 5 in panels B–D, *n* = 3 in panel (**F**)). ^#^
*p* < 0.05 and ^##^
*p* < 0.01 vs. the DAG group by unpaired Student’s *t*-test. O.D., optical density.

**Figure 4 cells-14-00991-f004:**
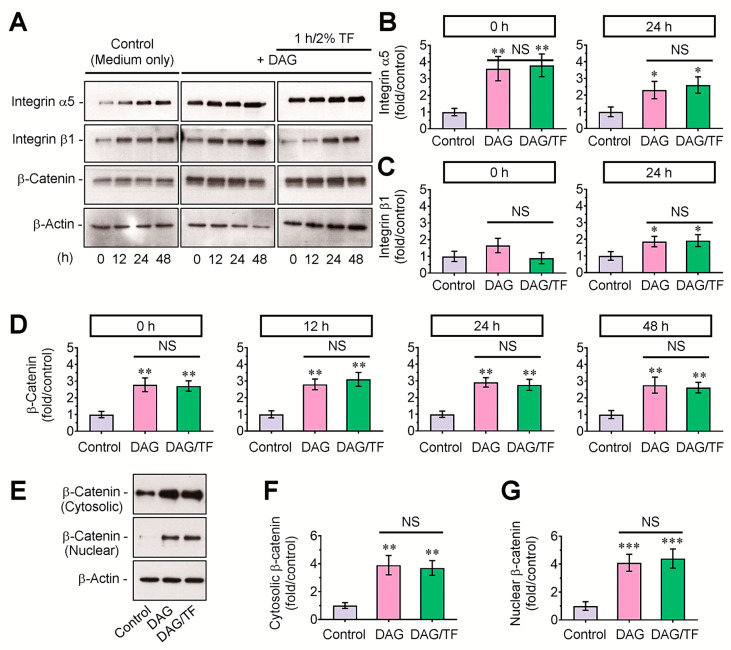
Tensile stress did not attenuate the DAG-stimulated induction of integrins and β-catenin in EBs. (**A**) Protein levels of integrins α5 and β1 and β-catenin in whole protein lysates of the EBs incubated in growth or osteogenic medium or in combination with tensile stress were analyzed by immunoblot assay at the indicated times. The band intensities of (**B**) integrin α5, (**C**) integrin β1, and (**D**) β-catenin among the groups were analyzed using the ImageJ program at each of the indicated times after normalizing their intensities to that of β-actin. (**E**) Levels of β-catenin in cytosolic and nucleic protein fractions of the control, DAG, and DAG/TF groups were determined by Western blotting at 24 h post-tension followed by densitometric analysis of β-catenin-specific band intensity in the (**F**) cytoplasm and (**G**) nucleus. Panels (**A**,**E**) show representative results from four different samples. * *p* < 0.05, ** *p* < 0.01, and *** *p* < 0.001 vs. the control group by unpaired Student’s *t*-test (*n* = 4). NS, not significant. O.D., optical density; TF, tensile force.

**Figure 5 cells-14-00991-f005:**
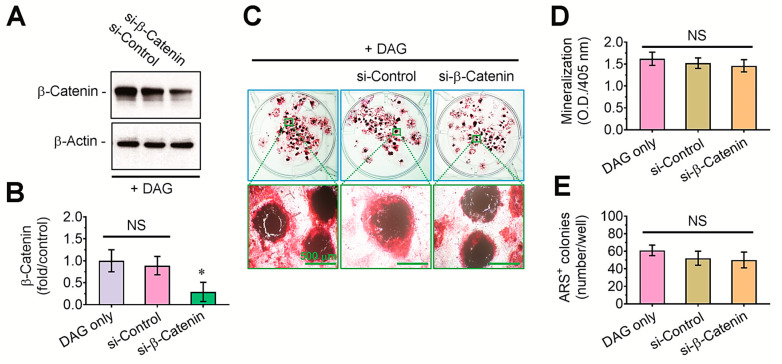
The knockdown of β-catenin by siRNA transfection did not affect DAG-stimulated mineralization of EBs. (**A**) The level of β-catenin in whole protein lysates of EBs was determined by immunoblot assay 48 h after the transfection with si-control or si-β-catenin. A representative result from four different samples is shown. (**B**) The fold change in β-catenin level was calculated after normalizing the band intensity to that of β-actin. (**C**) The EBs transfected with si-control or si-β-catenin were stained with ARS at 14 days post-incubation in DAG-supplied medium. (**D**) ARS-specific absorbance and (**E**) ARS-positive colony number were determined at the same post-incubation time. * *p* < 0.05 vs. the control group by unpaired Student’s *t*-test (*n* = 4). NS, not significant. O.D., optical density.

**Figure 6 cells-14-00991-f006:**
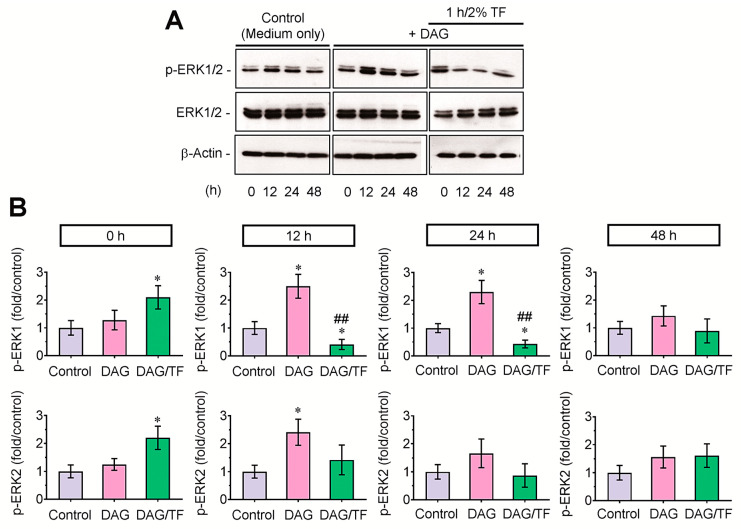
Tensile stress suppressed the phosphorylation of ERK in a time-dependent manner after the tensile force. (**A**) EBs were subjected to 2% tensile stress for 1 h, and at the indicated times of incubation, the levels of p-ERK1/2, ERK1/2, and β-actin were analyzed by Western blotting. A representative result from three different samples is shown. (**B**) Fold changes of p-ERK1 and 2 in these groups were compared by densitometric analysis after normalizing the band intensity to each of ERK1/2 at the indicated post-incubation times. * *p* < 0.05 vs. the control group by unpaired Student’s *t*-test (*n* = 3). ^##^
*p* < 0.01 vs. the DAG group by unpaired Student’s *t*-test (*n* = 3). TF, tensile force.

**Figure 7 cells-14-00991-f007:**
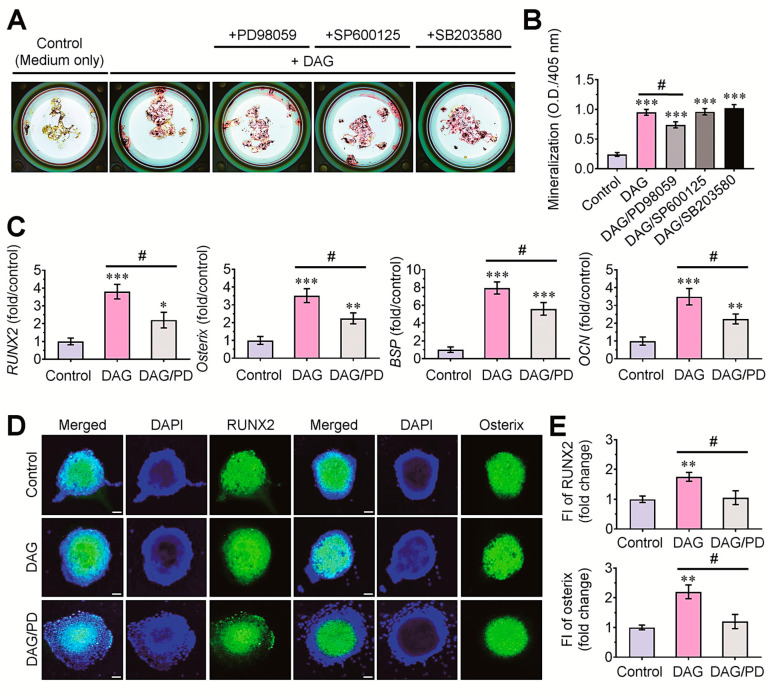
Pharmacological inhibitor specific to p-ERK, but not to p-JNK and p-p38 kinase, suppressed DAG-stimulated osteoblastic differentiation in EBs. (**A**) EBs were treated with each of the MAPK inhibitors (10 μM/inhibitor) and incubated in growth medium or DAG-contained medium for 14 days followed by ARS staining. (**B**) ARS dye-specific absorbance was measured at 405 nm. (**C**) Levels of *RUNX2*, *osterix*, *BSP*, and *OCN* were determined by real-time RT-PCR at 5 days post-treatment with PD98059 in the presence and absence of DAG. (**D**) Levels of RUNX2 and osterix proteins in the EBs were also evaluated by immunostaining assay at 5 days the same post-treatment. (**E**) Fluorescence intensity (FI) of RUNX2 and osterix proteins in the EBs was compared in relation to those cultures. * *p* < 0.05, ** *p* < 0.01, and *** *p* < 0.001 vs. the control group by unpaired Student’s *t*-test (*n* = 4). ^#^
*p* < 0.05 vs. the DAG group by unpaired Student’s *t*-test (*n* = 4). Bar = 50 μm. O.D., optical density.

**Figure 8 cells-14-00991-f008:**
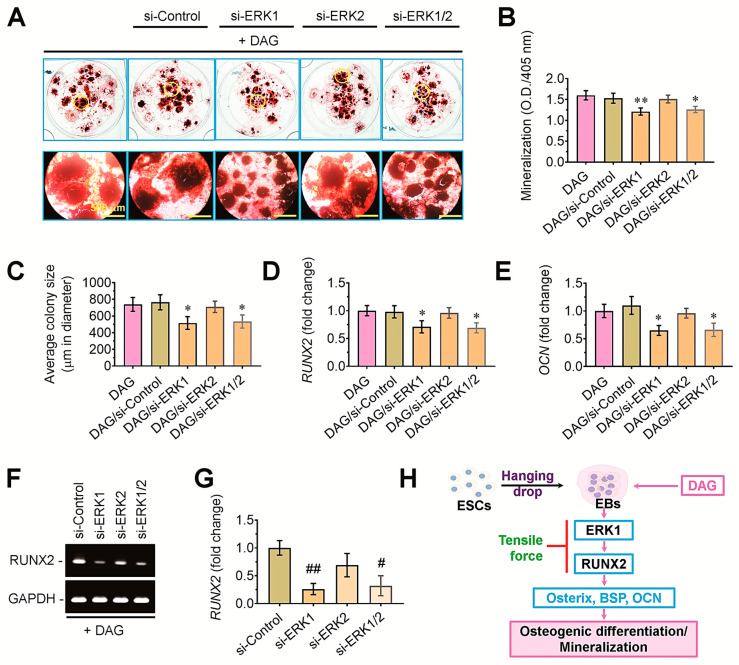
Transfection with si-ERK1 diminished DAG-stimulated mineralization of EBs. (**A**) EBs were transfected with si-control, si-ERK1, si-ERK2, or both si-ERK1 and 2 and incubated for an additional 14 days before the staining with ARS. EBs indicated in the yellow circles of upper panels were magnified as shown in the below panels. (**B**) ARS-specific absorbance in the groups was determined at 405 nm. (**C**) The average colony size of EBs was determined using ImageJ software (Ver. 1.51). The levels of (**D**) *RUNX2* and (**E**) *OCN* in the groups were measured by real-time PCR at 5 days post-transfection. (**F**) Agarose gel electrophoresis showing the expression pattern of *RUNX2* in relation to the siRNA transfection along with (**G**) the analysis of band intensity after normalizing the band to that of *GAPDH*. A representative result from three different samples is shown. (**H**) A schematic diagram indicating the signaling pathways involved in tension-mediated suppression of mineralization in DAG-stimulated EBs. * *p* < 0.05 and ** *p* < 0.01 vs. the DAG group by unpaired Student’s *t*-test (*n* = 4). ^#^
*p* < 0.05 and ^##^
*p* < 0.01 vs. the si-control group by unpaired Student’s *t*-test (*n* = 3).

## Data Availability

The data that support the findings of this study are available from the corresponding author upon reasonable request.

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
