# Peer review of "Tension Force Stress Downregulates the Expression of Osteogenic Markers and Mineralization in Embryonic Stem-Cell-Derived Embryoid Bodies"

_cells, 2025, doi:10.3390/cells14130991_

Round 1
Reviewer 1 Report
Comments and Suggestions for Authors
The manuscript “Tension Force Stress Downregulates the Expression of Osteo-2
genic Markers and Mineralization in Embryonic Stem Cell-De-3 rived Embryoid Bodies” is mostly clearly written, the experimental design is appropriate for testing the hypothesis and there is sufficient evidence to support the validity of the presented results. However, several important points require clarification to strengthen the analytical reliability and practical applicability of the study.
1) Introduction: Please explain which extracellular matrix provides the great advantage in GTR to enhance clarity.
2) The author concluded that “tensile stress suppresses DAG-stimulated osteogenic differentiation and mineralization of EBs, while only 2% cyclic tensile stress at 0.5 Hz was selected for conducting this experiment. Please briefly explain the rationale for selecting this specific condition.
3) Regarding Figures 3E, 4A, 4E, 5A, 7A and 8F, the number of replicates conducted for each experimental group is not specified. Please indicate the number of replicates or parallel tests parallel tests were performed for each dataset presented in the respective figures.
4) Please improve the quality and resolution of Figure 3A, 5C, 6A, and 8A.
Author Response
Reviewer 1
Comments and Suggestions for Authors
The manuscript “Tension Force Stress Downregulates the Expression of Osteo-2 genic Markers and Mineralization in Embryonic Stem Cell-De-3 rived Embryoid Bodies” is mostly clearly written, the experimental design is appropriate for testing the hypothesis and there is sufficient evidence to support the validity of the presented results. However, several important points require clarification to strengthen the analytical reliability and practical applicability of the study.
1) Introduction: Please explain which extracellular matrix provides the great advantage in GTR to enhance clarity.
â–ºAuthor response: Based on the reviewers’ comment, we provided additional explanation on the advantages of extracellular matrix in GTR into the Introduction section (page 2, lines 42~46).
2) The author concluded that “tensile stress suppresses DAG-stimulated osteogenic differentiation and mineralization of EBs, while only 2% cyclic tensile stress at 0.5 Hz was selected for conducting this experiment. Please briefly explain the rationale for selecting this specific condition.
â–ºAuthor response: A briefed explanation about the elongation ranges applied was already provided in the original submission in the Discussion section. In this revision, we additionally provided the reason why we selected 2% cyclic force as the tensile stress. The detail explanation was added into the Discussion section (page 13, lines 382~392).
3) Regarding Figures 3E, 4A, 4E, 5A, 7A and 8F, the number of replicates conducted for each experimental group is not specified. Please indicate the number of replicates or parallel tests parallel tests were performed for each dataset presented in the respective figures.
â–ºAuthor response: Actually, we performed Western blot analysis using three or more than three different samples, in which a representative result is shown in figures. To help the understating on the number of samples used, we added a brief explanation on the sample numbers on each of figure legends.
4) Please improve the quality and resolution of Figure 3A, 5C, 6A, and 8A.
â–ºAuthor response: In addition to the figures that the reviewer required to improve the resolution, we have tried to improve the quality of all figures in the revised manuscript.
Reviewer 2 Report
Comments and Suggestions for Authors
In the current manuscript, An and colleagues demonstrate Tensile stress induces downregulation of osteogenic markers and mineralization in embryoid bodies (EBs). The authors showed underlying mechanism of mineralization of EB via tensile stress, highlighting the involvement of ERK1 signaling pathway. Treatment of PD98059 reduced the mineralization of EBs and RUNX2 expression. In addition, knock-down of ERK1 but not ERK2 by siRNA significantly diminished the mineralization of EBs and RUNX2 expression. This will be very informative to identify the underlying mechanism of Tensile stress on EBs. There are several concerns that should be solved for publishing in Cells.
Major points
- The authors should consider reorganize the figure layout. Typically, schematic for experiments should be presented first to help readers understanding. Therefore, Figure 2A-D should be presented in Figure 1. Moreover, the order of Figure 6 and 7 should be considered. The current figure organization or structure makes it difficult to follow.
- Insufficient explanation for why only PD98059 showed a significant effect but not other ERK inhibitors. The authors should address this in discussion section.
- Reproducibility and specificity. Mouse ES cell line D3 was used in this study. However, the authors should validate these findings in other mouse ESCs or human ESCs for reproducibility and specificity, respectively.
- Additional experiments such as Calcium Quantification Assay should be performed for quantification or validation of osteogenic induction more accurately.
Minor points
- All abbreviations should be defined in the main text instead of Materials and Methods section (e.g. DAG). It makes readers confusing.
- There are many typos (e.g. DAD in figure 1 legend – line 214). The authors should carefully double check.
- Band of RUNX2 was not clear in Figure 3E. The quality of data can be improved if possible.
English should be improved, particularly by using more academic terminology to enhance the overall professionalism and clarity of your research.
Author Response
Reviewer 2
Comments and Suggestions for Authors
In the current manuscript, An and colleagues demonstrate Tensile stress induces downregulation of osteogenic markers and mineralization in embryoid bodies (EBs). The authors showed underlying mechanism of mineralization of EB via tensile stress, highlighting the involvement of ERK1 signaling pathway. Treatment of PD98059 reduced the mineralization of EBs and RUNX2 expression. In addition, knock-down of ERK1 but not ERK2 by siRNA significantly diminished the mineralization of EBs and RUNX2 expression. This will be very informative to identify the underlying mechanism of Tensile stress on EBs. There are several concerns that should be solved for publishing in Cells.
Major points
- The authors should consider reorganize the figure layout. Typically, schematic for experiments should be presented first to help readers understanding. Therefore, Figure 2A-D should be presented in Figure 1. Moreover, the order of Figure 6 and 7 should be considered. The current figure organization or structure makes it difficult to follow.
â–ºAuthor response: We thank the reviewer for providing meaningful comments. Based on the comments, we reorganized the order of figures 1, 2, 6, and 7. According to the changed orders, the Results section was also adequately revised.
- Insufficient explanation for why only PD98059 showed a significant effect but not other ERK inhibitors. The authors should address this in discussion section.
â–ºAuthor response: We thank the reviewer for pointing out the question in our study. Actually, we could not provide the exact reason by which the inhibition of ERK1, but not ERK2, shows a significant effect. However, our current findings can suggest the critical role of ERK-mediated signaling. As you know, PD98059 is the pharmacological inhibitor for ERK1/2, and thus the treatment of this inhibitor exerted such significant effect. To more understand the role of ERK, we performed si-transfection assay specific to ERK1 and ERK2, respectively. The results from si-ERK transfection supported the critical role of p-ERK1 rather than p-ERK2 in DAG-induced calcification of EBs. Collectively, our present findings suggest that tension-derived mechanical signals inhibit the activation of ERKs in DAG-supplied EBs, in which ERK1-mediated signaling tightly affects the osteogenic potential of the bodies. These explanations on the roles of ERK1 and ERK2 were also presented in the Discussion section (page 13, lines 421-428).
- Reproducibility and specificity. Mouse ES cell line D3 was used in this study. However, the authors should validate these findings in other mouse ESCs or human ESCs for reproducibility and specificity, respectively.
â–ºAuthor response: We thank the reviewer for providing meaningful comments. As the reviewer stated, it might be important to validate our findings using other stem cells such as other mouse ESCs or human-derived ESCs. However, we consider that such experiments to validate for reproducibility and specificity using other stem cells have to be performed as another project. We hope the reviewers’ understanding on the authors’response.
- Additional experiments such as Calcium Quantification Assay should be performed for quantification or validation of osteogenic induction more accurately.
â–ºAuthor response: We thank the reviewer for providing helpful comments. As the reviewer mentioned, the calcium quantification assay can improve the quality of this study on mineralization results. However, the alizarin red S (ARS) assay is also considered the gold standard for quantification of osteoblast mineralization and is thus widely used among scientists. In this study, we showed the results of ARS assay along with the dye quantification at all-related experiments to confirm the mineralized EBs. We consider that the current results can support the osteogenic induction of EBs. Frist of all, the main aims of this study were focused on the elucidation of molecular mechanisms by which tensile stress affect the mineralization of EBs. The authors hope the reviewers’ understanding on our response to the additional request. In addition, we replaced the word ‘calcification’ to ‘mineralization’ in the main text.
Minor points
- All abbreviations should be defined in the main text instead of Materials and Methods section (e.g. DAG). It makes readers confusing.
â–ºAuthor response: Thank you so much for providing helpful comment. However, we consider that the section of materials and methods belongs to the main text. In the case of the abbreviation, DAG, it is not suitable to define in the Introduction section. It might be shown in the part of methods. Instead, we have checked whether any abbreviations should be presented in the main text instead of Materials and Methods section if possible.
- There are many typos (e.g. DAD in figure 1 legend – line 214). The authors should carefully double check.
â–ºAuthor response: We are so sorry about such mistake. Before submission, we have entirely checked the contents of manuscript to avoid any typo-errors.
- Band of RUNX2 was not clear in Figure 3E. The quality of data can be improved if possible.
â–ºAuthor response: In addition to the Figure 3, the quality of all other figures was improved before resubmission.
Round 2
Reviewer 2 Report
Comments and Suggestions for Authors
In the current manuscript, An and colleagues demonstrate Tensile stress induces downregulation of osteogenic markers and mineralization in embryoid bodies (EBs). The authors addressed all the feedback thoroughly and improved the manuscript’s clarity and flow.
Major points
Minor points